

# Generalized hydrodynamics regime
# from the thermodynamic bootstrap program

Axel Cortés Cubero[1⋆] and Miłosz Panfil[2†]

**1** Institute for Theoretical Physics, University of Amsterdam,
Science Park 904, 1098 XH Amsterdam, The Netherlands
**2** Faculty of Physics, University of Warsaw, Pasteura 5, 02-093 Warsaw, Poland

⋆ a.cortescubero@uva.nl, † milosz.panfil@fuw.edu.pl

## Abstract

Within the generalized hydrodynamics (GHD) formalism for quantum integrable models, it is possible to compute simple expressions for a number of correlation functions at the Eulerian scale. Specializing to integrable relativistic field theories, we show the same correlators can be computed as a sum over form factors, the GHD regime corresponding to the leading contribution with one particle-hole pair on a finite energy-density background. The thermodynamic bootstrap program (TBP) formalism was recently introduced as an axiomatic approach to computing such finite-energy-density form factors for integrable field theories. We derive a new axiom within the TBP formalism from which we easily recover the predicted GHD Eulerian correlators. We also compute higher form factor contributions, with more particle-hole pairs, within the TBP, allowing for the computation of correlation functions in the diffusive, and beyond, GHD regimes. The two particle-hole form factors agree with expressions recently conjectured within the GHD.



# 1   Introduction

In this article, we explore the applications of the recently introduced Thermodynamic Bootstrap Program (TBP) [1] for correlation functions of integrable quantum field theories (IQFT) in the hydrodynamic regime. The thermodynamic bootstrap program is a set of axioms that strongly restrict form factors of physical operators between states of finite energy density, in IQFT's. One can then attempt to find exact expressions for these form factors, as self-consistent solutions of the set of axioms.

The TBP formalism is inspired by the standard integrable bootstrap program [2–4], which is used to compute exact form factors involving a finite number of particles on top of the vacuum. The TBP generalizes this formalism to the case where there is a finite number of particle and hole excitations on top of a thermodynamic background (itself consisting of an infinite number of background particles), rather than the vacuum. The background state can be, in principle, any eigenstate of IQFT characterized by a smooth filling function, e.g. the thermal state [5] or GGE state [6–8]. The thermodynamic form factors provides us with a fundamental ingredient for computation of the dynamic correlation functions in and out of the equilibrium.

Generalized Hydrodynamics (GHD) is an approach to study the dynamics of integrable models (including, but not limited to IQFT's) in spatially inhomogeneous setups. It is based on the idea that, given an inhomogeneous initial state, at late enough times, the spatial and temporal fluctuations are smooth enough, that they can be completely characterized by a set of hydrodynamical differential equations. It originated in [9, 10] as a way of solving the bipartite quench protocol, in which two thermodynamically different systems are joined. Since then, it was developed into a coherent framework [11–17] capable of describing more general inhomogeneous setups in different integrable models [9, 11, 18–26]. Recently, the predictions of GHD were also confirmed experimentally [27].

The regime where GHD is applicable is that where only spatial fluctuations of very long wavelength are relevant. As shown in the original work [10] and also elucidated more recently [28], the crucial ingredient in formulating the GHD is the knowledge of the small momentum limit of form factors of conserved densities and currents over a thermodynamic background. This creates a point of overlap between TBP and GHD which we aim to explore in this work. Specifically, we show that the predictions of TBP agree with the assumed structure of these form factors which was put forward while formulating the GHD.

In particular, it was shown in [13, 14, 29, 30], that the GHD formalism may be used to compute correlation functions of conserved charge and current densities in the hydrodynamical regime. Further, in [13] it was shown that for relativistic QFT's, Euler-scale correlators can be computed for general local operators. In the case of correlators of charge densities and currents, the same correlator can be computed as a sum over form factors over the thermodynamic background state. From the requirement that both expressions agree, one can derive from the GHD correlator, a simple prediction for low-momentum form factors of charge and current densities. We will show in this paper, that this prediction is easily recovered for IQFT's within the TBP formalism. Furthermore, we show that the general formula for any two local operators can be reproduced in the same way, by computing the one-particle-hole form factors.

Originally, the GHD was formulated at the Euler scales, where transport is generally ballistic. It was recently shown in [28, 31] that diffusive behavior can be re-introduced by considering form factors with higher numbers of particle-hole pairs on top of the background, than those included in the hydrodynamical regime. This results in a Navier-Stokes GHD formulation. An important ingredient in this formulation is an assumption on the form of the leading 2-particle-hole pair form factor. The conjectured form that was put forward in [31] was inspired by the results of [32] for the thermodynamic limit of the density operator form factor in the non-relativistic integrable Lieb-Liniger model. Specifically, in formulating the Navier-Stokes GHD, it was assumed that the leading diverging part of the 2 particle-hole form factor is universal for any local operator. In this paper, we will see that this GHD conjecture is also easily recovered within the TBP formalism, in the relativistic QFT setting. The TBP also allows us to study the low-momentum limit form factors with even higher numbers of particle-hole pairs, going beyond the Navier-Stokes GHD conjecture. It is worth mentioning that guided by kinematical arguments, an equivalent formula for the Navier-Stokes GHD term was proposed in [33].

The rest of the manuscript is organized in the following way. In Section 2 we briefly review the predictions from GHD for correlation functions at the Euler scale, as well as the conjectures for GHD diffusion corrections. In Section 3 we give a quick overview of the Thermodynamic Bootstrap program. In the following two Sections we evaluate the leading form factors within the TBP formalism. First, in Section 4 we evaluate the zero-momentum limit of the single particle-hole pair form factors for a generic local operator, which is the only necessary ingredient to compute Euler-scale correlators. Then, in Section 5, we consider form factors with multi particle-hole excitations. The case of two particle-hole pairs is relevant for the inclusion of diffusion to the hydrodynamic picture. Finally, in Section 6, we consider a simple application of these results to the computation of Euler-scale correlation functions.

## 2 Eulerian correlators from GHD and GHD with diffusion

The GHD formalism allows one to study the long-time evolution of systems with spatially inhomogeneous initial conditions. Such initial states can be described as GGE-like states, but with spatially varying chemical potentials. Following the notation of [13], local observables in an initial state at $t = 0$ are given by

$$\langle \mathcal{O} \rangle_{\text{ini}} = \frac{\text{Tr}\left(e^{-\int \mathrm{d}x \sum_i \beta^i(x) q^i(x)} \mathcal{O}\right)}{\text{Tr}\left(e^{-\int \mathrm{d}x \sum_i \beta^i(x) q^i(x)}\right)}, \tag{1}$$

where $q^i(x)$ are the charge densities, corresponding to the conserved charges of the integrable model, $Q^i = \int \mathrm{d}x\, q^i(x)$ and $\beta^i(x)$ are local chemical potentials. One is then interested in computing space- and time-dependent correlation functions

$$\langle \mathcal{O}(x_1, t_1) \dots \mathcal{O}_n(x_n, t_n) \rangle_{\text{ini}}, \tag{2}$$

where the time evolution of operators is given by $\mathcal{O}(x, t) = e^{\mathrm{i}tH} \mathcal{O}(x, 0) e^{-\mathrm{i}tH}$.

The Eulerian scale, where hydrodynamical equations are applicable, of such correlation functions, is given by considering late times and large separations between operators. Additionally, the Euler-scale correlators rely on the concept of *in-cell averaging*, which removes rapidly oscillating contributions. We define Eulerian correlators, as was done in [13], defining a mesoscopic fluid cell, denoted by $\mathcal{N}_\lambda(x, t)$, as a space-time region of size that scales as $\lambda^\nu$, for some $\nu_0 < \nu < 1$ around the scaled space-time point $\lambda x, \lambda t$, where $\lambda$ some large parameter. The parameter $\nu_0$ depends on the subleading corrections to the Euler scale, and if

those are of a diffusive character $\nu_0 = 1/2$. The fluid cell is then defined as the set of points $\mathcal{N}_\lambda(x,t) = \{(y,s) : \sqrt{(y-\lambda x)^2 + (s-\lambda t)^2} < \lambda^\nu\}$. The volume of the fluid cell is given by $|\mathcal{N}_\lambda| = \int_{\mathcal{N}_\lambda(x,t)} dy\,ds$. Eulerian correlation functions are then defined as

$$\langle \mathcal{O}_1(x_1,t_1)\ldots\mathcal{O}_N(x_N,t_N)\rangle_{\text{ini}}^{\text{Eulerian}} \tag{3}$$
$$= \lim_{\lambda\to\infty} \lambda^{N-1} \int_{\mathcal{N}_\lambda x_1,t_1} \frac{dy_1 ds_1}{|\mathcal{N}_\lambda|} \cdots \int_{\mathcal{N}_\lambda(x_N,t_N)} \frac{dy_N ds_N}{|\mathcal{N}_\lambda|} \langle \mathcal{O}_1(y_1,s_1)\ldots\mathcal{O}_N(y_N,s_N)\rangle_{\text{ini},\lambda}^{\text{connected}}.$$

At these scales, space and time dependence of the correlator can be captured by considering only the spatial and temporal dependence of chemical potentials in a local GGE. The time evolution of these chemical potentials is given by simple hydrodynamical differential equations. One central result of [13], is that at the Eulerian scale, correlation functions involving the charge density operators, $q_i(x)$, are given by

$$\left\langle q^i(x,0) \prod_k \mathcal{O}_k(x_k,t_k)\right\rangle_{\text{ini}}^{\text{Eulerian}} = -\frac{\delta}{\delta\beta^i(x)} \left\langle \prod_k \mathcal{O}_k(x_k,t_k)\right\rangle_{\text{ini}}^{\text{Eulerian}}. \tag{4}$$

In other words, at the Eulerian scale, one can insert charge-density operators into correlation functions, simply by taking functional derivatives with respect to the corresponding chemical potential. A similar result also holds for the current density operators, $j^i(x,t)$, corresponding to the conserved charges.

Using knowledge of one-point functions for charge and current densities, and equation (4), the following two-point functions (among other results) at the Eulerian scale where proposed in [13,34]

$$\langle q^i(x,t)q^j(0,0)\rangle_{\text{ini}}^{\text{Eulerian}} = \int d\theta\, \delta(x-v^{\text{eff}}(\theta)t)\rho_p(\theta)(1-\vartheta(\theta))q_{\text{dr}}^i(\theta)q_{\text{dr}}^j(\theta),$$
$$\langle j^i(x,t)q^j(0,0)\rangle_{\text{ini}}^{\text{Eulerian}} = \int d\theta\, \delta(x-v^{\text{eff}}(\theta)t)\rho_p(\theta)(1-\vartheta(\theta))v^{\text{eff}}(\theta)q_{\text{dr}}^i(\theta)q_{\text{dr}}^j(\theta), \tag{5}$$

where the functions $\rho_p(\theta)$ and $\vartheta(\theta)$ describe the distribution of the background particles in the GGE ensemble, $q_{\text{dr}}^i(\theta)$ is the dressed eigenvalue of the charge $Q^i$ for a particle of rapidity $\theta$ on top of the background, and $v^{\text{eff}}(\theta)$ is the velocity of a particle of rapidity $\theta$, dressed by the background. All these quantities will be defined in more detail in the following section. The results (5), as written, are valid for a homogeneous initial state, $\beta^i(x) = \beta^i$. This will suffice for the comparison with the TBP. The GHD provides a way to lift these expressions to an inhomogeneous setup. We refer again to [34] for details.

For integrable QFT's, within the GHD formalism, it was shown in [13] that Euler-scale correlation functions can be computed for arbitrary local operators. The procedure is to start from the known expression for the one-point correlation function of an operator in a GGE, $\langle \mathcal{O}(x,t)\rangle_{\text{ini}}$, which can be computed through the LeClair-Mussardo formula [35,36]. Then the charge density-generic operator Eulerian two-point function can be computed through Eq. (4). One is then able to extract from this correlator what is the contribution corresponding to each, the charge density, and the generic operator. Extracting and isolating the contribution from the generic operator, one can then write the general two point function (for a spatially homogeneous GGE state described by the filling function $\vartheta(\theta)$)

$$\langle \mathcal{O}_1(x,t)\mathcal{O}_2(0,0)\rangle_\vartheta^{\text{Eulerian}} - \langle \mathcal{O}_1\rangle_\vartheta\langle\mathcal{O}_2\rangle_\vartheta$$
$$= \int d\theta\, \delta(x-v^{\text{eff}}(\theta)t)\rho_p(\theta)(1-\vartheta(\theta))V^{\mathcal{O}_1}(\theta)V^{\mathcal{O}_2}(\theta), \tag{6}$$

where $V^{\mathcal{O}}(\theta)$ are operator-specific functions derived from the Leclair-Mussardo formula as

$$V^{\mathcal{O}}(\theta) = \sum_{k=0}^{\infty} \frac{1}{k!} \int \prod_{j=1}^{k} \left(\frac{d\theta_j}{2\pi}\vartheta(\theta_j)\right)(2\pi\rho_s(\theta))^{-1}f_c^{\mathcal{O}}(\theta_1,\ldots,\theta_k,\theta), \tag{7}$$

and $f_c^{\mathcal{O}}(\{\theta\})$ are the *connected* form factors of the operator, $\mathcal{O}$, which we will define more precisely in Section 4. In writing (6), we have used the effective velocity defined as $v^{\text{eff}}(\theta) = (E')^{\text{dr}}(\theta)/(p')^{\text{dr}}(\theta)$, where the "dressing" procedure for the energy and momentum are explained in more detail in the next section. The integral in (6) localizes to countable number of contributions given by $\theta_*(\xi)$ which are solutions of the equation $v^{\text{eff}}(\theta) = \xi$, where $\xi \equiv x/t$,

$$\langle \mathcal{O}_1(x,t)\mathcal{O}_2(0,0)\rangle_{\vartheta}^{\text{Eulerian}} - \langle \mathcal{O}_1\rangle_{\vartheta}\langle \mathcal{O}_2\rangle_{\vartheta} = t^{-1}\sum_{\theta \in \theta_*(\xi)} \frac{\rho_p(\theta)(1-\vartheta(\theta))}{|(v^{\text{eff}})'(\theta)|} V^{\mathcal{O}_1}(\theta)V^{\mathcal{O}_2}(\theta). \quad (8)$$

We will show explicitly in Section 6 how expression (8) can be fully recovered within the TBP formalism, by including only form factors with one particle-hole pair with the same rapidity on top of the thermodynamic background.

It was recently shown in [28,31] that diffusive behavior can be reintroduced into the GHD formalism by introducing into the hydrodynamic equations a term proportional to the diffusion matrix, defined as

$$(\mathcal{D}C)_{ij} = \int dt \left[\int dx \, \langle j^i(x,t)j^j(0,0)\rangle_{\vartheta}^{\text{connected}} - \left(\lim_{t'\to\infty}\int dx \langle j^i(x,t')j^j(0,0)\rangle_{\vartheta}^{\text{connected}}\right)\right]. \quad (9)$$

If one were to compute such diffusion matrix in terms of a form factor expansion, the first non-trivial contribution would come from the low-momentum limit of the two particle-hole pair expansion (whereas the Eulerian scale involves only one particle-hole pair). It has been argued that the form of this diffusion matrix at this leading order should be

$$\begin{aligned}(\mathcal{D}C)_{ij} &= \int \frac{d\theta d\alpha}{2}\rho_p(\theta)f(\theta)\rho_p(\alpha)f(\alpha)|v_{\text{dr}}(\theta)-v_{\text{dr}}(\alpha)| \\ &\times \left(\frac{q_{\text{dr}}^i(\theta)T^{\text{dr}}(\theta,\alpha)}{\rho_s(\theta)} - \frac{q_{\text{dr}}^i(\alpha)T^{\text{dr}}(\alpha,\theta)}{\rho_s(\alpha)}\right) \\ &\times \left(\frac{q_{\text{dr}}^j(\theta)T^{\text{dr}}(\theta,\alpha)}{\rho_s(\theta)} - \frac{q_{\text{dr}}^j(\alpha)T^{\text{dr}}(\alpha,\theta)}{\rho_s(\alpha)}\right),\end{aligned} \quad (10)$$

where $\rho_s(\theta)$ is the density of states describing the thermodynamic background, and $T^{\text{dr}}(\theta,\alpha)$ is related to the dressed scattering phase (we will define more clearly both functions in the following sections), and $f(\theta)$ is a statistical factor, depending on whether particles are bosonic or fermionic. This diffusion matrix follows from the conjecture on the the form factor structure put forward in Ref. [31]. We will show that the conjectured form factor structure can be derived explicitly for IQFT's in the TBP formalism, validating thus the diffusion matrix (10).

## 3 Thermodynamic bootstrap program

In this section we provide a short summary of the Thermodynamic Bootstrap Program. For a detailed presentation, we refer to [1]. The aim of the TBP is to provide a formalism for computing form factors in IQFTs at finite energy density. The existence of an infinite number of local conserved charges in IQFT's results in the absence of creation and annihilation processes and factorization of any $n$-body scattering process into a product of 2-body scattering processes [4]. The IQFT's are characterized by the elastic 2-body scattering matrix $S(\theta_1 - \theta_2)$, where the rapidity is defined as the parametrization of the on-shell energy and momentum of particles, $E = m\cosh(\theta)$, $p = m\sinh(\theta)$. For the remainder of this paper, we will consider

IQFT's with only one species of particles of mass $m$, for simplicity of presentation, but our results could be generalized for theories with a richer spectrum.

The states at finite energy density, such as the thermal states, are described by specifying the filling function $\vartheta(\theta)$, describing the probability of finding a particle of rapidity $\theta$ on the state. Due to the interactive nature of the theory, the physical density of the particles, $\rho_p(\theta)$, depends on the presence of other particles and follows from the integral equation

$$\rho_s(\theta) = m\cosh(\theta) + \int d\theta'\, T(\theta, \theta')\rho_p(\theta'), \quad \rho_p(\theta) = \vartheta(\theta)\rho_s(\theta), \tag{11}$$

where $\rho_s(\theta)$ is the total density of particles and $T(\theta, \theta')$ is the scattering kernel related to the $S$-matrix [1],

$$T(\theta, \theta') = \frac{1}{2\pi}\frac{\partial}{\partial\theta}\delta(\theta - \theta'), \qquad \delta(\theta) = -i\log(-S(\theta)). \tag{12}$$

The energy density (for system size $L$) of a state is

$$\frac{E}{L} = \int d\theta\, \rho_p(\theta)\cosh(\theta), \tag{13}$$

and expectation values of other conserved charges are

$$\langle Q^i \rangle_\vartheta = \frac{\langle\vartheta|Q^i|\vartheta\rangle}{\langle\vartheta|\vartheta\rangle} = L\int d\theta\, \rho_p(\theta)q^i(\theta), \tag{14}$$

where $q^i(\theta)$ is the charge eigenvalue on a one-particle state, $Q^i|\theta\rangle = q^i(\theta)|\theta\rangle$.

We define a finite-energy density state, $|\vartheta\rangle$, as an eigenstate with an extensive number of particles, whose rapidities are distributed according to the filling fraction $\vartheta(\theta)$. We then consider excitations on top of this thermodynamic background, by adding or removing a finite number of particles, characterized by a set of rapidities $\{\theta_j\}_{j=1}^n$. We denote such excited states by $|\vartheta; \theta_1, \ldots, \theta_n\rangle$. Again, due to the interactive nature of the theory, the excited state not only has extra particles, but also the distribution of the background particles $\vartheta(\theta)$ is slightly shifted. This shift is described by the back-flow function

$$F(\theta|\{\theta\}) = \sum_{j=1}^n F(\theta|\theta_j), \tag{15}$$

$$F(\theta|\theta_j) = \frac{1}{2\pi}\delta(\theta - \theta_j) + \int d\theta'\, T(\theta, \theta')\vartheta(\theta')F(\theta'|\theta_j), \tag{16}$$

which dictates the amount by which the rapidity of a background particle, originally located at $\theta$, is shifted by the introduction of a set of excitations with rapidities $\{\theta\}$. In the limit of the low momentum particle-hole excitation the back-flow is effectively captured by dressed differential scattering phase

$$T^{dr}(\theta, \theta_1) = T(\theta, \theta_1) + \int d\theta'\vartheta(\theta')T(\theta', \theta)T^{dr}(\theta, \theta_1). \tag{17}$$

The effect of the shift on the expectation values of conserved charges can be encapsulated by the dresing procedure. The expectation values of a conserved charge $Q^i$ on a state $|\vartheta; \theta_1, \ldots, \theta_n\rangle$ is

$$\frac{\langle\vartheta; \theta_1, \ldots, \theta_n|Q^i|\vartheta; \theta_1, \ldots, \theta_n\rangle}{\langle\vartheta; \theta_1, \ldots, \theta_n|\vartheta; \theta_1, \ldots, \theta_n\rangle} = \langle Q^i\rangle_\vartheta + q^i_{eff}(\theta_1) + \cdots + q^i_{eff}(\theta_n), \tag{18}$$

---

[1]To agree with the GHD notation we use here a slightly different notation than in [1]. Namely, the integral equations are written in terms of the kernel $T(\theta, \theta')$, instead the standard IQFT notation uses $\varphi(\theta, \theta') = 2\pi T(\theta, \theta')$.

where we define the effective charge as the bare charge of a particle, $q(\theta)$ plus the shift in the value of the background's charge, induced by the presence of the new particle, or

$$q_{\text{eff}}(\theta) = q(\theta) - \int d\theta' \vartheta(\theta') q'(\theta') F(\theta'|\theta). \tag{19}$$

We will later use proper names, $k(\theta)$ and $\omega(\theta)$ to refer to the effective momentum and energy of an excitation. The derivative of the effective charge can be shown to satisfy

$$(q_{\text{eff}}^i)'(\theta) = (q^i)'(\theta) + \int d\theta' \, T(\theta, \theta') \vartheta(\theta') (q_{\text{eff}}^i)'(\theta'). \tag{20}$$

From this integral equation (20), it is useful to define the "dressing" procedure, for a general function $g(\theta)$, producing a dressed function as

$$g_{\text{dr}}(\theta) = g(\theta) + \int d\theta' \, T(\theta, \theta') \vartheta(\theta') g_{\text{dr}}(\theta'), \tag{21}$$

or written in operatorial form

$$g(\theta) = ((1 - T\vartheta) \, g_{\text{dr}})(\theta). \tag{22}$$

Its formal solution is

$$g_{\text{dr}}(\theta) = \left((1 - T\vartheta)^{-1} \, g\right)(\theta), \tag{23}$$

with the dressing operator $(1 - T\vartheta)^{-1}$. For example, the dressed momentum and energy, follow from the dressing of the single particle expectation values $m \sinh \theta$ and $m \cosh \theta$ respectively. Another example of the dressing procedure in action is the integral equation (17).

Form factors of local operators within the standard integrable bootstrap program are defined as the functions,

$$f^{\mathcal{O}}(\theta_1, \dots, \theta_n) \equiv \langle 0 | \mathcal{O} | \theta_1, \dots, \theta_n \rangle. \tag{24}$$

The bootstrap program consists in formulating a set of axioms that these functions satisfy, and which strongly constrain the form factors. The end goal is to have enough constraints such that the functions $f^{\mathcal{O}}(\theta_1, \dots, \theta_n)$ can be recovered analytically as the minimal function that consistently satisfies all the axioms [2–4].

The main goal of the TBP is to generalize the concept of form factor axioms to the case where particle and hole excitations are on top of a (generalized) thermodynamic background, rather than on top of the vacuum, as in the standard bootstrap program. We define now a form factor of a local operator $\mathcal{O}(x)$ as the function

$$f_{\vartheta}^{\mathcal{O}}(\theta_1, \dots, \theta_n) = \frac{\langle \vartheta | \mathcal{O}(0) | \vartheta; \theta_1, \dots, \theta_n \rangle}{\langle \vartheta | \vartheta \rangle}. \tag{25}$$

We can interpret it as an n-particle form factor that is dressed and modified by the presence of the background described by the distribution $\vartheta(\theta)$. These form factors need to be properly defined by first considering a finite volume expression with a finite number of particles, then taking the thermodynamic limit. We will confront this issue in more detail in the next section. In relativistic field theories, introducing a hole excitation (or removing a particle excitation) of rapidity $\theta$ is equivalent to introducing a particle with rapidity $\theta + \pi i$, such that the expression (25) can describe both, particle and hole excitations. The shift in rapidity $\theta + \pi i$ can also be interpreted in terms of crossing symmetry, as adding a particle of rapidity $\theta$ in the bra, instead of the ket state.

These form factors enter the expression for the two-point correlation function [1]

$$\frac{\langle\vartheta|\mathcal{O}(x,t)\mathcal{O}(0,0)|\vartheta\rangle}{\langle\vartheta|\vartheta\rangle} = \sum_{n=0} \sum_{\sigma_i=\pm1} \left(\prod_{k=1}^{n} \fint_{-\infty}^{\infty} \frac{\mathrm{d}\theta}{2\pi} \vartheta_{\sigma_k}(\theta_k)\right)$$
$$\times f_\vartheta^{\mathcal{O}}(\theta_1,\ldots,\theta_n)_{\sigma_1,\ldots,\sigma_n} \left(f_\vartheta^{\mathcal{O}'}(\theta_1,\ldots,\theta_n)_{\sigma_1,\ldots,\sigma_n}\right)^*$$
$$\times \exp\left(\mathrm{i}x \sum_{k=1}^{n} \sigma_k k(\theta_k) - \mathrm{i}t \sum_{k=1}^{n} \sigma_k \omega(\theta_k)\right), \tag{26}$$

where we define the filling fractions,

$$\vartheta_{-1}(\theta_k) = \vartheta(\theta_k), \qquad \vartheta_{+1}(\theta_k) = \frac{\rho_h(\theta)}{\rho_p(\theta)}\vartheta(\theta_k) \tag{27}$$

and

$$f_\vartheta^{\mathcal{O}}(\theta_1,\ldots,\theta_n)_{\sigma_1,\ldots,\sigma_n} = f_\vartheta^{\mathcal{O}}(\theta_1 + i\pi\delta_{\sigma_1,-1},\ldots,\theta_n + i\pi\delta_{\sigma_n,-1}). \tag{28}$$

Functions $k(\theta)$ and $\omega(\theta)$ are the effective energy and momentum, respectively, of a particle of rapidity $\theta$. The two-point function is written as an expansion in terms of the dressed form factors, with particle and hole excitations on top of the background, $\vartheta$. We use $\sigma$ to distinguish particles ($\sigma = 1$) from holes ($\sigma = -1$). Moreover, behind the $\fint$ symbol there is a prescription for the integration which should be performed along a line shifted by $i\epsilon$ above the real axis and with the residues from the double annihilation poles subtracted.

The formula (26) corrects some issues with the previously conjectured Leclair-Mussardo formula [37]. In particular, it involves the new form factors that have been dressed by the thermodynamic background, instead of the vacuum form factors used in [37]. The formula (26) incorporates also a precise prescription to deal with the singularities arising from the form factors, whereas in the proposal of [37] no regularization procedure was proposed. An alternate regularization procedure for the Leclair Mussardo formula was recently proposed in [38]. The formula proposed in [38] can in practice only be computed as a low temperature expansion which sets it far from the GHD regime. Finally, we note that within the TBP the axioms are formulated in the microcanonical approach using the concept of a representatitve state. An approach based on formulating axioms for the genuine Gibbs ensemble, put forward in [39] for Ising field theory, turned out to be difficult to generalize to interacting IQFT's.

In [1] we postulated that form factors (25) obey a set of axioms. We will not list all of these here, but only show some of the most relevant ones to our discussion, which are

- Scattering axiom

$$f_\vartheta^{\mathcal{O}}(\theta_1,\ldots,\theta_i,\theta_{i+1},\ldots,\theta_n) = S(\theta_i - \theta_{i+1})f_\vartheta^{\mathcal{O}}(\theta_1,\ldots,\theta_{i+1},\theta_i,\ldots,\theta_n). \tag{29}$$

- Periodicity axiom

$$f_\vartheta^{\mathcal{O}}(\theta_1,\ldots,\theta_n) = R_\vartheta(\theta_n|\theta_1,\ldots,\theta_n)f_\vartheta^{\mathcal{O}}(\theta_n + 2\pi i, \theta_1,\ldots,\theta_{n-1}). \tag{30}$$

- Annihilation pole axiom

$$-i\mathrm{Res}_{\theta_1=\theta_2} f_\vartheta^{\mathcal{O}}(\theta_1,\theta_2,\ldots,\theta_n) =$$
$$[1 - R_\vartheta(\theta_2|\theta_3,\ldots,\theta_n)S(\theta_2 - \theta_3) \times \cdots \times S(\theta_2 - \theta_n)]f_\vartheta^{\mathcal{O}}(\theta_3,\ldots,\theta_n). \tag{31}$$

- Spatial and temporal translation

$$
\begin{aligned}
f_\vartheta^{\mathcal{O}}(x,t)(\theta_1,\dots,\theta_n) &\equiv \frac{\langle \vartheta | \mathcal{O}(x,t) | \vartheta; \theta_1,\dots,\theta_2 \rangle}{\langle \vartheta | \vartheta \rangle} \\
&= e^{ixm\sum_{i=1}^{n} k(\theta_i) - itm \sum_{i=1}^{n} \omega(\theta_i)} f_\vartheta^{\mathcal{O}}(\theta_1,\dots,\vartheta).
\end{aligned}
\tag{32}
$$

Here, $R_\vartheta(\theta | \theta_1,\dots,\theta_n)$ is related to the back-flow function through

$$
R_\vartheta(\theta | \theta_1,\dots,\theta_n) = \prod_{j=1}^{n} R_\vartheta(\theta | \theta_j),
\tag{33}
$$

$$
R_\vartheta(\theta | \theta_j) = \exp\left(2\pi i F(\theta | \theta_j) - i\delta(\theta - \theta_j)\right).
\tag{34}
$$

The presented axioms generalize the vacuum axioms [2,3] and reduce to them in the $\vartheta(\theta) \to 0$ limit. In this limit, $R_0(\theta | \theta_j) = 1$.

We will see later, in Section 6, that when computing correlation functions of local operators at the Euler scale, we only need to consider the leading contribution to the expansion (26). This contribution is given by considering only form factors with one particle-hole pair excitation on top of it, in the zero-momentum limit (in the limit where the rapidity of the particle and the hole coincide, such that the momentum and energy of the pair vanishes). That is, the only ingredient we need to compute Euler-scale correlators are form factors of the form $\lim_{\kappa \to 0} f_\vartheta^{\mathcal{O}}(\theta + \pi i, \theta + \kappa)$. In the next section we evaluate and find an analytic expression for such form factors.

The annihilation pole axiom will be of particular importance to our discussion in the Section 5. This axiom controls the form factor in the region where a particle-hole pair has rapidities very close to each other, in the presence of other excitations. This region is important for obtaining diffusive corrections to the GHD dynamics.

Finally, we note that the form factors in the GHD convention [31] differ from our convention by appearance of extra factors $\rho_s(\theta_j)$. For a form factor with $n$ particle-hole excitations

$$
f_\vartheta^{\mathcal{O}}(\theta_1^+,\dots,\theta_n^+, \theta_1^- + i\pi,\dots,\theta_n^- + i\pi) = \frac{\langle \vartheta | \mathcal{O} | \vartheta; \{\theta_j^+, \theta_j^-\} \rangle_{\text{GHD}}}{\sqrt{\prod_{j=1}^{n} 2\pi \rho_s(\theta_j^+) 2\pi \rho_s(\theta_j^-)}}.
\tag{35}
$$

## 4 Single particle-hole form factors at low momentum

In this section, we are interested in the evaluation of the form factor

$$
\lim_{\kappa \to 0} f_\vartheta^{\mathcal{O}}(\theta + \pi i, \theta + \kappa).
\tag{36}
$$

In particular, we will obtain an expression for this form factor in terms of the standard, zero-background form factors.

Our first step is to regularize the form factor by studying its leading contributions at a large but finite volume, $L$. At finite volume, the particle rapidities are quantized, given by the solutions, $\{\tilde{\theta}\}$ of the Bethe equations [4]

$$
Q_k(\theta_1,\dots,\theta_n) \equiv Lm \sinh\theta_k + \sum_{l \neq k} \delta(\theta_k - \theta_l) = 2\pi I_k,
\tag{37}
$$

for each value of $k \in [1, n]$. The numbers $\{I\}$ are a set of integers (or half integers, depending on the particle statistics), for each particle. The Bethe equations then serve as a mapping from a set of quantum numbers $\{I\}$ into a set of of rapidities $\{\theta\}$.

At finite volume, it is then convenient to express form factors in the basis of the set of integers $\{I\}$. Following the results of [40–42], the finite-volume form factors can be expressed in terms of the infinite-volume form factors, as

$$\langle I'_1, \ldots, I'_m | O(0) | I_1, \ldots, I_n \rangle = \frac{f^{\mathcal{O}}(\theta'_1 + \pi i, \ldots, \theta'_m + \pi i, \theta_1, \ldots, \theta_n)}{\sqrt{\rho_m(\theta'_1, \ldots, \theta'_m) \rho_n(\theta_1, \ldots, \theta_n)}} + e^{-\mu L}, \tag{38}$$

where this expression is exact up to all powers of $L$, the only corrections being exponentially suppressed (which will not be important for our discussion, so we will ignore them). The functions $\rho_n(\{\theta\})$ are given by the determinant of the Jacobian of the transformation from integers, $\{I\}$, to rapidities, $\{\theta\}$,

$$\begin{aligned} J_{kl}(\theta_1, \ldots, \theta_n) &\equiv \frac{\partial}{\partial \theta_l} Q_k(\theta_1, \ldots, \theta_n), \\ \rho_n(\theta_1, \ldots, \theta_n) &\equiv |\text{Det} \, J(\theta_1, \ldots, \theta_n)|. \end{aligned} \tag{39}$$

The finite-volume regularization of the form factor (36) is then

$$f^{\mathcal{O}}_{\vartheta}(\theta + \pi i, \theta + \kappa) = \lim_{L \to \infty} \frac{f^{\mathcal{O}}(\theta_n + \pi i, \ldots, \theta_1 + \pi i, \theta_1 + \kappa_1, \ldots, \theta_n - \kappa_n, \theta + \pi i, \theta + \kappa)}{\rho_n(\theta_1, \ldots, \theta_n)}, \tag{40}$$

where, as derived in Appendix A, in the limit of a small particle-hole excitation,

$$\kappa_j = -\frac{T^{\text{dr}}_L(\theta_j, \theta)}{L \rho_{s,L}(\theta_j)} \kappa. \tag{41}$$

Functions $T^{\text{dr}}_L(\theta, \theta')$ and $\rho_{s,L}(\theta)$ are finite-volume versions of the standard thermodynamic functions $T^{\text{dr}}(\theta, \theta')$ and $\rho_s(\theta)$. Their defining equations are given in Appendix A. The set of rapidities $\theta_1, \ldots, \theta_n$, with $n \sim L$ are chosen such that in the thermodynamic limit they are distributed according to occupation number $\vartheta(\theta)$,

$$\lim_{L,n \to \infty} \sqrt{\rho_n(\theta_1, \ldots, \theta_n)} |I_1, \ldots, I_n\rangle = |\vartheta\rangle. \tag{42}$$

Note that, in accordance with the definition of the thermodynamic form factors (25), the right hand side of (40) is normalized with respect to the background state.

The ordering of the rapidities in (40) follows the convention of the thermodynamic form factors, in which excitations appear after the background rapidities. In the limit $\kappa \to 0$ the ordering of (40) can be brought to the standard IQFT ordering in which $\theta + \pi i$ appears first. To achieve this one has to scatter the corresponding particle with all other particles. In doing so the back-flow accumulates, however the back-flow is of order $\kappa$ and therefore does not contribute to the leading order when $\kappa \to 0$. Therefore

$$f^{\mathcal{O}}_{\vartheta}(\theta + \pi i, \theta + \kappa) =$$
$$\lim_{L \to \infty} \frac{f^{\mathcal{O}}(\theta + \pi i, \theta_n + \pi i, \ldots, \theta_1 + \pi i, \theta_1 + \kappa_1, \ldots, \theta_n - \kappa_n, \theta + \kappa)}{\rho_n(\theta_1, \ldots, \theta_n)} \times (1 + \mathcal{O}(\kappa)). \tag{43}$$

This expression, together with the relation (41) between $\kappa_j$ and $\kappa$, is the starting point of our analysis.

To better understand expression (43), it is necessary to examine more closely the singularity structure of form factors. Following the annihilation-pole axiom [2,3], form factors are singular whenever the rapidity of an incoming and outgoing particle approach each other. For

an almost diagonal form factor (with all the particles in the incoming state approaching those in the outgoing state), we can express this singular structure as

$$f^{\mathcal{O}}(\theta_n + \pi i, \ldots, \theta_1 + \pi i, \theta_1 + \kappa_1, \ldots, \theta_n + \kappa_n) = \frac{1}{\kappa_1 \ldots \kappa_n} \sum_{\{i_k\}_n=1}^{n} a_{i_1,\ldots,i_n} \kappa_{i_1} \ldots \kappa_{i_n}, \tag{44}$$

for small $\{\kappa_i\}$, where the functions $a_{i_1,\ldots,i_m}$ are all finite and symmetric in the indices. The summation is

$$\sum_{\{i_k\}_n=1}^{n} (\cdots) = \sum_{i_1=1}^{n} \cdots \sum_{i_n=1}^{n} (\cdots). \tag{45}$$

It is convenient to extract certain finite values related to the diagonal form factors (44). First, one notices that the expression (44) is completely finite if all the regulators are taken to zero simultaneously, $\kappa_i = \kappa$, for all $i$. From this one can define the *symmetric* form factor, as

$$\begin{aligned} f_s^{\mathcal{O}}(\theta_1, \ldots, \theta_n) &\equiv \lim_{\kappa \to 0} f^{\mathcal{O}}(\theta_n + \pi i, \ldots, \theta_1 + \pi i, \theta_1 + \kappa, \ldots, \theta_n + \kappa) \\ &= \sum_{\{i_k\}_n=1}^{n} a_{i_1,\ldots,i_n}. \end{aligned} \tag{46}$$

Another useful quantity is the *connected* form factor, defined as the finite (non diverging) part of the form factor as *any* of the $\kappa_i$ is individually taken to zero. The connected form factor is then given by

$$f_c^{\mathcal{O}}(\theta_1, \ldots, \theta_n) \equiv n! \, a_{1,\ldots,n}. \tag{47}$$

Factor $n!$ appears from summing over permutations of $\kappa_i$.

In [43] it was shown, that the almost diagonal form factor of the form

$$f^{\mathcal{O}}(\theta_n + \pi i, \ldots, \theta_1 + \pi i, \theta_1 + \epsilon_1, \ldots, \theta_n + \epsilon_n), \tag{48}$$

in the limit of small $\epsilon_i$, can be represented by connected form factors with different number of particles. The result is best understood in terms of sum over graphs which we now introduce following [43]. Let $G$ be a set of the directed graphs $G_i$ with $n$ vertices and with the following properties

1. $G_i$ is tree-like,

2. For each vertex there is at most one outgoing edge.

We denote $E_{jk}$ an edge going from $j$ to $k$. Then the diagonal form-factor is expressed as a sum over all graphs in $G$. With each node of the graph $G_i$ we associate a rapidity. Choice of graph $G_i$ divides the set of all the rapidities in two sets $\{\theta\} = \{\theta_+\} \cup \{\theta_-\}$, where we associate $\{\theta_+\}$ to the nodes without outgoing arrow and $\{\theta_-\}$ to the nodes with an outgoing arrow. Each graph contributes then

- a factor $f_c^{\mathcal{O}}(\{\theta_+\})$

- for each edge $E_{jk}$, the form factor is multiplied by $2\pi(\epsilon_k/\epsilon_j)T(\theta_j - \theta_k)$.

We now apply this general result of [43] to the particle-hole form factor (36) expressed as a thermodynamic limit of the finite volume form factor (43). That is we consider $2(n+1)$ particle diagonal form factor with rapidities $\{\theta_j\}_{j=1}^{n}$ and $\theta$ and $\epsilon_i = \kappa_i$ with an exception that $\epsilon_{n+1} = \kappa$. For small $\kappa$ we have that $\kappa_j = \alpha_j \kappa$ where $\alpha_i \equiv T_L^{\mathrm{dr}}(\theta_i, \theta)/L\rho_{s,L}(\theta_i)$ for $i \in [1, n]$ and

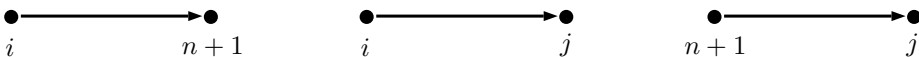

Figure 1: The three types of graphs contributing to $C^1$. We picture only the 2 connected nodes of each class, with all other nodes disconnected. The pictures correspond to the contributions $C^{1a}$, $C^{1b}$ and $C^{1c}$, respectively.

$\alpha_{n+1} = 1$. This leads to the new rule, that for each edge $E_{jk}$ in the graph $G_i$ the form factor is multiplied by $2\pi(\alpha_k/\alpha_j)T(\theta_j - \theta_k)$.

In the finite system this is the final result for the particle-hole form factor. The structure of terms contributing to the form factor changes in the thermodynamic limit. The reason for this is that all the factors $\alpha_j$ are of order $1/L$ with an exception of $\alpha_{n+1} = 1$. Therefore the contribution from the graphs were the $n+1$ node has ingoing edges are potentially more important in the thermodynamic limit. To analyse this carefully we first order the summation over graphs by the number of edges they contain. We denote $C^m$ contribution to the form-factor from graphs with $m$ edges.

There is a single graph with $n+1$ nodes and without edges. Therefore

$$C^0 \equiv \frac{f_c^{\mathcal{O}}(\theta_1, \ldots, \theta_n, \theta)}{\rho_n(\theta_1, \ldots, \theta_n)}. \tag{49}$$

Graphs with a single edge can be further divided in 3 groups depending on whether the $(n+1)$-th node has an incoming edge, there is no edge connected to it, there is an outgoing edge. We call the 3 contributions $C^{1a}$, $C^{1b}$ and $C^{1c}$ respectively. The graphs corresponding to each contribution are depicted in Fig. 1. We have

$$C^{1a} = 2\pi \sum_{i=1}^n \frac{f_c^{\mathcal{O}}(\{\theta\}_{n+1}, \hat{\theta}_i)}{\rho_n(\theta_1, \ldots, \theta_n)} \frac{1}{\alpha_i} T(\theta_i - \theta),$$

$$C^{1b} = 2\pi \sum_{\substack{i,j=1 \\ i \neq j}}^n \frac{f_c^{\mathcal{O}}(\{\theta\}_{n+1}, \hat{\theta}_i)}{\rho_n(\theta_1, \ldots, \theta_n)} \frac{\alpha_j}{\alpha_i} T(\theta_i - \theta_j),$$

$$C^{1c} = 2\pi \sum_{i=1}^n \frac{f_c^{\mathcal{O}}(\{\theta\}_{n+1}, \hat{\theta}_{n+1})}{\rho_n(\theta_1, \ldots, \theta_n)} \alpha_i T(\theta - \theta_i), \tag{50}$$

where we used that $\alpha_{n+1} = 1$, and we have introduced the notation $(\{\theta\}_{n+1}, \hat{\theta}_i)$ to represent the set of $n+1$ rapidities, but where the rapidity $\theta_i$ has been removed. Contributions $C^{1a}$ and $C^{1b}$ are of order $L^2$, whereas $C^{1c}$ is of order $L^0$ and can be neglected. In the thermodynamic limit, the leading contribution is thus

$$C^1 =_{\text{th}} 2\pi \sum_{i=1}^n \frac{f_c^{\mathcal{O}}(\{\theta\}_{n+1}, \hat{\theta}_i)}{\rho_n(\theta_1, \ldots, \theta_n)} \frac{1}{\alpha_i} \left( T(\theta_i - \theta) + \sum_{\substack{j=1 \\ j \neq i}}^n \alpha_j T(\theta_i - \theta_j) \right)$$

$$= 2\pi L \sum_{i=1}^n \frac{f_c^{\mathcal{O}}(\{\theta\}_{n+1}, \hat{\theta}_i) \rho_s(\theta_i)}{\rho_n(\theta_1, \ldots, \theta_n)}, \tag{51}$$

where we have introduced the equivalence symbol, $=_{\text{th}}$, to mean that the two expressions agree in the thermodynamic limit. In writing (51) we used the integral equation for the dressed scattering $T^{\text{dr}}(\theta_i, \theta)$, Eq. (21) in its finite-volume version from Appendix A.

We can continue this way, computing the leading contributions $C^m$, coming from graphs with $m$ edges. We analyze in the detail the case of $m = 2$ in the Appendix B. The result is

$$C^2 =_{\text{th}} \frac{1}{2} \sum_{\substack{i,j=1 \\ i \neq j}}^{n} \frac{f_c^{\mathcal{O}}(\{\theta\}_{n+1}, \hat{\theta}_i, \hat{\theta}_j) \tilde{\rho}_n(\theta_i, \theta_j)}{\rho_n(\theta_1, \ldots, \theta_n)}. \tag{52}$$

Similarly one can continue calculating $C^m$ for each value of $m$, from which we can extrapolate the general expression

$$\lim_{\kappa \to 0} f_\vartheta^{\mathcal{O}}(\theta + \pi \mathrm{i}, \theta + \kappa) = \lim_{L,n \to \infty} \sum_{m=1}^{n} C^m = \lim_{L,n \to \infty} \sum_{\{\theta_+\} \cup \{\theta_-\} = \{\theta\}} \frac{f_c^{\mathcal{O}}(\{\theta_+\}, \theta) \tilde{\rho}_n(\{\theta_-\})}{\rho_n(\{\theta_-\} \cup \{\theta_+\})}, \tag{53}$$

where we are summing over all the possible bipartitions of the set of rapidities $\{\theta\} = \{\theta_1, \ldots, \theta_n\} = \{\theta_-\} \cup \{\theta_+\}$. We have also $\tilde{\rho}_n(\{\theta_-\})$ as the determinant of the submatrix which is defined in terms of the Jacobian $J_{kl}$, by selecting only the rows and colums with the indices $k, l$ corresponding to particles in the subset $\{\theta_-\}$.

We want to further evaluate the expression (53), term by term in the thermodynamic limit. This calculation will be similar to the derivation of the Leclair-Mussardo formula shown in Ref. [44]. To do this, it is convenient to start with the term corresponding to $\{\theta_+\} = \emptyset$ and $\{\theta_-\} = \{\theta\}$. Then we will evaluate the next term where $\{\theta_+\}$ contains only one particle, and so on. We can thus express (53) as a series

$$\lim_{\kappa \to 0} f_\vartheta^{\mathcal{O}}(\theta + \pi \mathrm{i}, \theta + \kappa) = D^0 + D^1 + D^2 + \ldots, \tag{54}$$

where $D^i$ are the contributions in (53) where $\{\theta_+\}$ contains $i$ particles. We will assume the sum over bipartitions and the thermodynamic limit can be exchanged, such that we will take the thermodynamic limit of each term individually.

First, we consider the term $D^0$, which corresponds simply to the form factor

$$D^0 = f_c^{\mathcal{O}}(\theta). \tag{55}$$

For the next term, $D^1$, we consider $\{\theta_-\}$ with one particle, and sum over all these one particle contributions, we find

$$D^1 = \int \frac{d\theta_1}{2\pi} \vartheta(\theta_1) f_c^{\mathcal{O}}(\theta_1, \theta), \tag{56}$$

where we used that [44]

$$\frac{\tilde{\rho}_n(\theta_1, \ldots, \theta_{i-1}, \theta_{i+1}, \ldots, \theta_n)}{\rho_n(\theta_1, \ldots, \theta_n)} = \frac{1}{2\pi L \rho_s(\theta_i)}, \tag{57}$$

and expressed the discrete sum over particles as an integral over rapidities,

$$\lim_{L,n \to \infty} \sum_i f(\theta_i) = L \int d\theta \, \rho_p(\theta) f(\theta). \tag{58}$$

It is then straightforward to continue evaluating, term by term, contributions with more particles in $\{\theta_+\}$. This produces the series,

$$\lim_{\kappa \to 0} f_\vartheta^{\mathcal{O}}(\theta + \pi \mathrm{i}, \theta + \kappa) = 2\pi \rho_s(\theta) V^{\mathcal{O}}(\theta) = \sum_{k=0}^{\infty} \frac{1}{k!} \int_{S^{\times k}} \prod_{j=1}^{k} \left( \frac{d\theta_j}{2\pi} \vartheta(\theta_j) \right) f_c^{\mathcal{O}}(\theta_1, \ldots, \theta_k, \theta). \tag{59}$$

This formula is the main result of this section. As we will see in Section 6, it reproduces the result for correlators at the Eulerian scale, derived from the GHD formalism [13], while we reinstate that absolutely no knowledge or assumptions from GHD were used in our derivation.

The scattering and periodicity axioms, Eqs. (29), (30) can be generally solved in terms of a minimal form factor times an ambiguous periodic factor. The two-particle (related to the one particle-hole pair by analytic continuation) minimal factor is a unique function (up to a normalization constant) which satisfies both of these axioms, and has no zeros or poles in the "physical strip", defined by $Im[\theta_i] \in [0, 2\pi]$. The full form factor can be written as the minimal form factor multiplied by a periodic function $K_{\vartheta}^{\mathcal{O}}(\theta_1, \theta_2) = K_{\vartheta}^{\mathcal{O}}(\theta_1 + \pi i, \theta_2) = K_{\vartheta}^{\mathcal{O}}(\theta_1, \theta_2 + \pi i)$, which encodes the properties of the operator $\mathcal{O}$. This function presents an inherent ambiguity of the form factor bootstrap approach, which is also present in the standard form factor bootstrap program, without thermodynamic background [4]. A typical approach to fix the ambiguity is to choose the minimal function which includes all the necessary poles and zeros expected by the particular structure of the operator, and which is consistent with the UV properties of correlation functions (which at short distances should approach the results from a CFT fixed point).

In our first paper [1], the same approach was adopted towards the computation of the finite temperature form factors of vertex operators in the sinh-Gordon model. Once the analytic structure of poles and zeros was fixed, we made the minimal assumption that the remaining ambiguity factor is only an overall normalization constant, which we were able to fix in terms of known quantities. Instead, our new result provides a powerful condition to constraint the ambiguous function $K_{\vartheta}^{\mathcal{O}}(\theta_1, \theta_2)$. This function can be now chosen as the one that satisfies the new condition Eq. (59), this way fixing the ambiguity.

It is important to note that Eq. (59) is not derived from our previous set of axioms, and we derive it directly from knowledge of the finite-volume regularization of standard, zero temperature form factors. It can therefore be treated as a new independent axiom that complements the set of axioms derived in [1].

We point out the similarity of our result (59) to dressed form factors in the crossed channel previously considered in [45]. We recall that due to Lorentz invariance, the partition function of a (1+1)-d QFT at finite temperature and infinite volume is equivalent to that of a finite-volume theory at zero temperature. The form factors studied in [45] concern particle excitations of the finite-volume Hamiltonian at zero temperature, while our result involves the infinite-volume particle excitations, dressed by a finite temperature (or GGE background). It seems a very intriguing coincidence that the dressing procedure for form factors in both channels seems to match, up to some imaginary shifts in the particle rapidities. The agreement of the dressing procedure for form factors in both channels was already proposed in Ref. [46] where the analytic structure of thermal form factors for the free fermionic theory was studied in detail. Our result suggests this correspondence between finite volume and finite temperature form factors may be carried through to interacting field theories as well. This connection is a subject that merits a deeper study in the future.

## 5 Two and more particle-hole pairs

Computing the low-momentum limit of form factors with a higher number of pairs is now straightforward and the formula reduces to the single particle-hole pair form factors through the annihilation pole axiom (31).

We consider the simplest generalization, to the case of two particle-hole pairs at low momentum. For a generic form factor of an operator, $\mathcal{O}$, in the presence of a background, the

annihilation pole axiom requires that at the leading order in $\kappa_1$

$$f_{\vartheta}^{\mathcal{O}}(\theta_1 + \pi i + \kappa_1, \theta_1, \theta_2 + \pi i + \kappa_2, \theta_2)$$
$$\sim \frac{i}{\kappa_1}[1 - R(\theta_1|\theta_2 + \pi i + \kappa_2, \theta_2)S(\theta_1 - \theta_2 - \pi i - \kappa_2)S(\theta_1 - \theta_2)]$$
$$\times f_{\vartheta}^{\mathcal{O}}(\theta_2 + \pi i + \kappa_2, \theta_2), \tag{60}$$

with function $R$ defined in (34). The factor in brackets in the right-hand side vanishes in the $\kappa_2 \to 0$ limit. The leading contribution to the 2-particle-hole pairs form factor at this point comes at order $\mathcal{O}(\kappa_2/\kappa_1)$, which is finite, as long as $\kappa_2 \sim \kappa_1$ as we take the $\kappa_{1,2} \to 0$ limit. Therefore, taking first small $\kappa_1$ limit and then small $\kappa_2$ limit we find

$$f_{\vartheta}^{\mathcal{O}}(\theta_1 + \pi i + \kappa_1, \theta_1, \theta_2 + \pi i + \kappa_2, \theta_2) \sim 2\pi \frac{\kappa_2}{\kappa_1}\frac{\partial F(\theta_1|\theta_2)}{\partial \theta_2} f_{\vartheta}^{\mathcal{O}}(\theta_2 + \pi i + \kappa_2, \theta_2). \tag{61}$$

The derivative of the back-flow is related to the dressed scattering phase shift (17),

$$\frac{\partial F(\theta_1|\theta_2)}{\partial \theta_2} = -T^{\mathrm{dr}}(\theta_1, \theta_2). \tag{62}$$

Hence the non-analytic contribution to the form factor is

$$f_{\vartheta}^{\mathcal{O}}(\theta_1 + \pi i + \kappa_1, \theta_1, \theta_2 + \pi i + \kappa_2, \theta_2) \sim -2\pi \frac{\kappa_2}{\kappa_1}T^{\mathrm{dr}}(\theta_1, \theta_2)f_{\vartheta}^{\mathcal{O}}(\theta_2 + \pi i + \kappa_2, \theta_2). \tag{63}$$

We can evaluate further the structure of poles of the 2 particle-hole-pair form factor, by considering different order of the limits (first with $\kappa_2$ and then with $\kappa_1$) and different permutations of particles and holes that may annihilate with each other. Through the annihilation-pole axiom, we can fully determine the non-analytic part of the form factors as a function of 4 rapidities. We find

$$\boxed{\begin{aligned} f_{\vartheta}^{\mathcal{O}}(\theta_1, \theta_2, \theta_3, \theta_4) = -2\pi \sum_{\sigma \in P_4} S_{\sigma}(\theta_1, \ldots, \theta_4)\frac{\theta_{\sigma_3} - \theta_{\sigma_4} - i\pi}{\theta_{\sigma_1} - \theta_{\sigma_1} - i\pi}T^{\mathrm{dr}}(\theta_{\sigma_2}, \theta_{\sigma_4})f_{\vartheta}^{\mathcal{O}}(\theta_{\sigma_3}, \theta_{\sigma_4}) \\ + \text{regular}, \end{aligned}} \tag{64}$$

where $S_{\sigma}(\theta_1, \ldots, \theta_4)$ is a product over the $S$-matrices obtained from representing permutation $\sigma$ as a product of adjacent transpositions and then writing the $S$-matrix for every transposition. For example, for permutation $\sigma = (1423) = (23)(34)$,

$$S_{(1423)}(\theta_1, \ldots, \theta_4) = S(\theta_2 - \theta_4)S(\theta_3 - \theta_4). \tag{65}$$

The pole structure of the form factor (64) agrees with the conjectured form factors in [31]. It has been shown in [31], that if the 2-particle-hole-pair form factors of current density operators are of the form (64), this leads to a diffusion matrix of the form (10).

The structure carries on for higher number of particle-hole pairs. For example, for 3 particle-hole pairs, taking the limit first with $\kappa_1$, then with $\kappa_2$ and finally with $\kappa_3$ we find the non-analytic contribution to be

$$f_{\vartheta}^{\mathcal{O}}(\theta_1 + \pi i + \kappa_1, \theta_1, \theta_2 + \pi i + \kappa_2, \theta_2, \theta_3 + \pi i + \kappa_3, \theta_3)$$
$$\sim (2\pi)^2 \left[ \frac{\kappa_3}{\kappa_1}T^{\mathrm{dr}}(\theta_1, \theta_2)T^{\mathrm{dr}}(\theta_2, \theta_3) + \frac{\kappa_3^2}{\kappa_1 \kappa_2}T^{\mathrm{dr}}(\theta_1, \theta_3)T^{\mathrm{dr}}(\theta_2, \theta_3) \right]$$
$$\times f_{\vartheta}^{\mathcal{O}}(\theta_3 + \pi i + \kappa_3, \theta_3). \tag{66}$$

Considering the limit in different orders and mixing the particle and holes pairs we can capture the whole singularity structure of the form factor in a formula generalizing (64).

# 6 Euler-scale correlators from particle-hole form factors

In this section we will show how our results for the single particle-hole pair form factors from Section 4 yield the Euler-scale correlation functions predicted from the GHD formalism in Ref. [13]. We consider the two-point function

$$C^{\mathcal{O}_1\mathcal{O}_2}(\xi,t) \equiv \frac{\langle\vartheta|\mathcal{O}_1(\xi t,t)\mathcal{O}_2(0,0)|\vartheta\rangle}{\langle\vartheta|\vartheta\rangle} - \frac{\langle\vartheta|\mathcal{O}_1|\vartheta\rangle}{\langle\vartheta|\vartheta\rangle}\frac{\langle\vartheta|\mathcal{O}_2|\vartheta\rangle}{\langle\vartheta|\vartheta\rangle} \tag{67}$$

in the large $t$ limit, there are two types of contributions to this correlator. One coming from particle-hole excitations with a leading single particle-hole pair term; ultimately, this contribution yields the GHD correlator. However, there are also contributions from un-equal number of particles and holes excitations. These contributions will be of an oscillatory character and therefore vanish under the in-cell averaging. We start by considering the leading one-particle-hole pair contribution.

Within the TBP formalism, the general expression for the two-point function is given by Eq. (26). Taking the large-times limit, however, means that we can focus only on low-energy excitations, as intermediate states with a high number of particles and holes are suppressed at long times/distances. At the Euler scale, we therefore consider only the lowest-lying excitation: a single particle-hole pair, whose contribution is

$$\begin{aligned} C^{\mathcal{O}_1\mathcal{O}_2}(\xi,t) &= \int \frac{d\theta_1}{2\pi}\frac{d\theta_2}{2\pi}\vartheta(\theta_1)(1-\vartheta(\theta_2))f_\vartheta^{\mathcal{O}_1}(\theta_1,\theta_2+\pi\mathrm{i})\left(f_\vartheta^{\mathcal{O}_2}(\theta_1,\theta_2+\pi\mathrm{i})\right)^* \\ &\quad \times \exp\left\{\mathrm{i}t\left[\xi(k(\theta_1)-k(\theta_2))-(\omega(\theta_1)-\omega(\theta_2))\right]\right\}. \end{aligned} \tag{68}$$

In the large $t$ limit, this integral can be readily computed using a two-dimensional stationary phase approximation. Defining the two-dimensional vector $\vec{\theta} = (\theta_1,\theta_2)$, and the functions in this space

$$\begin{aligned} g(\vec{\theta}) &= \frac{\vartheta(\theta_1)(1-\vartheta(\theta_2))}{(2\pi)^2}f_\vartheta^{\mathcal{O}_1}(\theta_1,\theta_2+\pi\mathrm{i})\left(f_\vartheta^{\mathcal{O}_2}(\theta_1,\theta_2+\pi\mathrm{i})\right)^*, \\ h(\vec{\theta}) &= \xi[k(\theta_1)-k(\theta_2)]-\omega(\theta_1)+\omega(\theta_2), \end{aligned} \tag{69}$$

the stationary phase approximation yields

$$C^{\mathcal{O}_1\mathcal{O}_2}(\xi,t) = \frac{2\pi}{t}g(\vec{\theta}_0)\left|\det H[h(\vec{\theta}_0)]\right|^{-1/2}\exp\left[\mathrm{i}t\,h(\vec{\theta}_0)+\mathrm{i}\frac{1}{4}\sigma_H\right], \tag{70}$$

where $H[h(\vec{\theta})]$ is the Hessian of the function $h(\vec{\theta})$, *i.e.* the two-by-two matrix

$$H_{ij}[h(\vec{\theta})] = \frac{\partial^2 h(\vec{\theta})}{\partial\theta_i\partial\theta_j}, \tag{71}$$

$\vec{\theta}_0$ are the set of critical points that satisfy $\nabla h(\theta)|_{\vec{\theta}=\vec{\theta}_0} = 0$ and $\sigma_H$ is the signature of the Hessian (number of positive eigenvalues minus number of negative eigenvalues) evaluated at $\vec{\theta} = \vec{\theta}_0$.

The Hessian can then be expressed as the matrix,

$$\begin{aligned} &H\;[h(\vec{\theta})] \\ &= \begin{pmatrix} k''(\theta_1)(\xi-v^{\mathrm{eff}}(\theta_1))-k'(\theta_1)\left(v^{\mathrm{eff}}\right)'(\theta_1) & 0 \\ 0 & -k''(\theta_2)(\xi-v^{\mathrm{eff}}(\theta_2))+k'(\theta_2)\left(v^{\mathrm{eff}}\right)'(\theta_2) \end{pmatrix}. \end{aligned} \tag{72}$$

The critical points are also easily computed, giving $\vec{\theta}_0 = (\theta_*(\xi), \theta_*(\xi))$, where we recall from Section 2 the definition $v^{\text{eff}}[\theta_*(\xi)] = \xi$. With this information, it is easy to verify that $\sigma_H = 0$, and that

$$\left| \det H[h(\vec{\theta}_0)] \right|^{1/2} = k'(\theta_*)(v^{\text{eff}})'(\theta_*) = 2\pi\rho_s(\theta_*)(v^{\text{eff}})'(\theta_*). \tag{73}$$

At the critical points $\vec{\theta}_0$, both rapidities are equal, $\theta_1 = \theta_2 = \theta_*(\xi)$, therefore we can directly use the results for the particle-hole form factors from Eq. (59). Lastly, to match more closely the notation of [13], we notice we can write

$$g(\vec{\theta}_0) = (2\pi)^2 \rho_s(\theta_*) \rho_p(\theta_*)(1 - \vartheta(\theta_*)) V^{\mathcal{O}_1}(\theta_*) V^{\mathcal{O}_2}(\theta_*). \tag{74}$$

Putting all these ingredients together into (70) we finally arrive at the correlator

$$C^{\mathcal{O}_1\mathcal{O}_2}(\xi, t) = t^{-1} \sum_{\theta \in \theta_*(\xi)} \frac{\rho_p(\theta)(1 - \vartheta(\theta))}{|(v^{\text{eff}})'(\theta)|} V^{\mathcal{O}_1}(\theta) V^{\mathcal{O}_2}(\theta) + \mathcal{O}(1/t^2), \tag{75}$$

with $V^{\mathcal{O}}(\theta)$ given by Eq. (59). Now considering the in-cell averaging described in (4), we see this leaves the expression (75) invariant, because it depends smoothly on the positon, $x, t$, so that averaging about a region $\mathcal{N}_\lambda(x, t)$ will not modify it. We have thus reproduced precisely the GHD prediction [13] for the Euler-scale correlation function (6) by performing a form factor expansion, and keeping only the leading contribution, from one particle-hole pair excitations. The subleading terms in (75) appear from corrections to the saddle-point approximations for one particle-hole excitations and from contributions with higher number of particle-hole excitations as we will now illustrate.

The presented analysis for the single particle-hole excitation can be repeated for $m$ particle-hole pairs, and the resulting contribution at large times behaves like $t^{-m}$. This confirms that the Euler scale correlations are determined by the single particle-hole processes. For an illustration of a general behaviour, we consider $m = 2$ case, for which the contribution to the correlation function is

$$C^{\mathcal{O}_1\mathcal{O}_2}_{2\text{ph}}(\xi, t) = \fint \mathrm{d}\vec{\theta} \, g(\vec{\theta}) \exp\left(ith(\vec{\theta})\right), \tag{76}$$

where $\vec{\theta}$ is now 4-dimensional vector and $g(\vec{\theta})$ and $h(\vec{\theta})$ are straightforward generalizations of (69) to the case of 2 particle-hole excitations. Behind the integration, there is a certain regularization scheme due to the presence of double poles in $g(\vec{\theta})$ whenever a position of a hole coincides with a position of a particle. The details of the regularization are however not important for the present analysis. At large time $t$ the integration in $C^{\mathcal{O}_1\mathcal{O}_2}_{2\text{ph}}(\xi, t)$ is appropriate for the stationary phase approximation logic. The critical point of $h(\vec{\theta})$ is $\vec{\theta}_0 = (\theta_*(\xi), \theta_*(\xi), \theta_*(\xi), \theta_*(\xi))$ again with $v^{\text{eff}}[\theta_*(\xi)] = \xi$. We change now the integration variables to $\vec{x} = \sqrt{t}(\vec{\theta} - \vec{\theta}_0)$ to find

$$C^{\mathcal{O}_1\mathcal{O}_2}_{2\text{ph}}(\xi, t) = \frac{1}{t^2} \fint \mathrm{d}\vec{x} \, g\left(\vec{\theta}_0 + \frac{x_i}{\sqrt{t}}\right) \exp\left(\frac{i}{2} \sum_{i,j=1}^{2} x_i H_{ij}[h(\vec{\theta}_0)] x_j\right). \tag{77}$$

The subtlety lies now in the fact that stationary phase approximation localises the integrand to the double pole of $g(\vec{\theta})$. The double pole comes from the non-analytic structure of the form-factor as displayed in (64). We observe that the singular part is controlled by the ratio of rapidities differences and the rest of the form-factor is regular and finite when rapidities coincide. The ratio of rapidities differences does not depend on $t$ when evaluated at $\theta_*(\xi) + x_i/\sqrt{t}$, the argument of the integrand in $C^{\mathcal{O}_1\mathcal{O}_2}_{2\text{ph}}(\xi, t)$. Therefore, the integrand is an analytic function

in $1/\sqrt{t}$ with a leading term of order 1. This implies that indeed the contribution to the correlation function from 2 particle-hole excitations vanishes at large times as $t^{-2}$. Straightforward generalization of this argument to larger number of particle-hole excitations yields the leading contribution from $m$ particle-hole excitations to be of order $t^{-m}$.

The form factor expansion also generates oscillatory terms that will vanish under the in-cell-averaging procedure. These terms are generated by considering form factors with an un-equal number of particles and holes. The first such term is given by the one-particle (or one-hole) form factor. This would give a contribution to the two point function,

$$C_{1\text{p}}^{\mathcal{O}_1 \mathcal{O}_2}(\xi, t) = \int \frac{d\theta}{2\pi} (1 - \vartheta(\theta)) f_\vartheta^{\mathcal{O}_1}(\theta) \left( f_\vartheta^{\mathcal{O}_2}(\theta) \right)^* e^{it[\xi k(\theta) - \omega(\theta)]}. \tag{78}$$

Again the leading contribution at late times can be computed via a stationary phase approximation, yielding[2],

$$C_{1p}^{\mathcal{O}_1 \mathcal{O}_2}(\xi, t) \approx \frac{1}{\sqrt{t}} \sum_{\theta \in \theta_*(\xi)} (1 - \vartheta(\theta)) \sqrt{\frac{1}{\rho_s(\theta)(v^{\text{eff}})'(\theta)}} f_\vartheta^{\mathcal{O}_1}(\theta) \left( f_\vartheta^{\mathcal{O}_2}(\theta) \right)^* e^{i(\xi k(\theta) - \omega(\theta))t \pm i\frac{\pi}{4}}. \tag{79}$$

This contribution, even though it decays with a power of $t^{-1/2}$, is oscillatory in time, which means that when we perform the in-cell averaging defined in (4), the contribution will vanish. We point out that these contributions may be related to the oscillatory terms that were observed in the classical integrable field theories in [47]

We conclude with a remark that the two-particle-hole pairs form factors, from Eq. (64) could be now used to derive the diffusion matrix (10). Since this calculation has already been shown in Ref. [31], we will not reproduce it here, but refer the reader to the original paper.

# 7 Conclusions

We have shown that the predictions of the Thermodynamic Bootstrap Program in the hydrodynamic regime agree with those of Generalized Hydrodynamics.

We have shown that the form factor for a generic local operator in an integrable QFT, with one particle-hole pair on top of a thermodynamic background, can be computed, in the limit when the two rapidites approach each other, in terms of a LeClair-Mussardo-like expansion over connected form factors. These form factors can be used to calculate two-point functions of operators, and we have shown that to compute Euler-scale correlators, it is sufficient to include the one-particle-hole form factors. We have shown that the correlation function thus derived exactly matches the GHD prediction from Ref. [13].

Additionally, we computed the pole structure of form factors with 2 particle-hole pairs, corresponding to the Navier-Stokes GHD regime, through the TBP annihilation pole axiom, accounting for dressing by the thermodynamic background. Our result agrees with the conjectured 2 particle-hole pairs form factor structure that leads to appropriate diffusive transport.

An interesting question for the future is about establishing a systematic gradient expansion of the GHD. For this, the necessary ingredients are higher particle-hole pairs form factors of conserved densities and currents. We have shown that the Thermodynamic Bootstrap program automatically generates such form factors and might be of value in further developing GHD in that direction.

The results of this work also serve as a non-trivial check of the correctness of the Thermodynamic Bootstrap Program. In particular, the results for the 2-particle-hole pair form factors

---

[2]The sign of $i\pi/4$ term is determined by the sign of the second derivative of the oscillatory factor evaluated at the stationary point, see Eq. (70).

suggest that the annihilation-pole axiom, dressed by the thermodynamic background, as proposed in [1], is indeed correct. It would be interesting to further develop computations within the TBP, to provide other non-trivial checks against the wealth of results from the GHD literature.

## Acknowledgments

The authors thank Jacopo De Nardis, Benjamin Doyon and Alvise Bastianello for valuable insights and comments on the early version of this work. ACC acknowledges the support from the European Research Council under ERC Advanced grant 743032 DYNAMINT, as well as support from the European Union's Horizon 2020 research and innovation programme under the Marie Skłodowska-Curie grant agreement 75009 at early stages of this project. MP acknowledges the support from the National Science Centre under SONATA grant 2018/31/D/ST3/03588.

## A  Back-flow of the rapidities in the presence of a small particle-hole excitation

In this appendix we consider the change to the rapidities of the background state induced by a presence of a particle-hole excitation of a small momentum. The rapidities of the background state in a finite system fulfil the Bethe equations

$$mL \sinh \theta_j = 2\pi I_j - \sum_{\substack{k=1 \\ k \neq j}}^{n} \delta(\theta_j - \theta_k), \qquad j = 1, \ldots, n. \tag{80}$$

Creating a particle-hole excitations amounts to adding a particle $\bar{\theta}$ and anti-particle $\theta$. Because of the coupled nature of the Bethe equations this leads to a change in all the other rapidities as well. We denote the set of modified rapidities by $\{\bar{\theta}\}$ and consider a difference between the Bethe equations for rapidities in the excited state and the background state

$$mL \left( \sinh \bar{\theta}_j - \sinh \theta_j \right) = -\sum_{\substack{k=1 \\ k \neq j}}^{n} \left( \delta(\bar{\theta}_j - \bar{\theta}_k) - \delta(\theta_j - \theta_k) \right) - \delta(\bar{\theta}_j - \bar{\theta}) + \delta(\bar{\theta}_j - \theta), \tag{81}$$

$$j = 1, \ldots, n.$$

A particle-hole excitation is small if $\bar{\theta} = \theta + \kappa$ with small parameter $\kappa$. As a consequence, the difference $\bar{\theta}_j - \theta_j$ is also small. Expanding the equality in this difference we find in the leading order

$$mL \left( \bar{\theta}_j - \theta_j \right) \cosh \theta_j = -2\pi \sum_{\substack{k=1 \\ k \neq j}}^{n} \left( \bar{\theta}_j - \theta_j - \bar{\theta}_k + \theta_k \right) T(\theta_j - \theta_k) - 2\pi\kappa \, T(\theta_j - \theta_n). \tag{82}$$

Reorganizing the sum leads to

$$L \left( \bar{\theta}_j - \theta_j \right) \left( \frac{m \cosh \theta_j}{2\pi} + \frac{1}{L} \sum_{\substack{k=1 \\ k \neq j}}^{n-1} \phi(\theta_j - \theta_k) \right) = \sum_{\substack{k=1 \\ k \neq j}}^{n-1} \left( \bar{\theta}_k - \theta_k \right) T(\theta_j - \theta_k) - \kappa \, T(\theta_j - \theta_n). \tag{83}$$

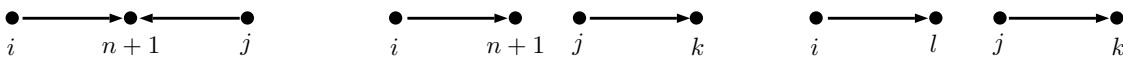

Figure 2: The three types of graphs contributing to $C^2$ (at leading order at large $L$), with all the disconnected nodes not pictured. The pictures correspond to the contributions $C^{2a}$, $C^{2b}$ and $C^{2c}$, respectively.

We define now two functions

$$\rho_L(\theta) = \frac{m\cos\theta}{2\pi} + \frac{1}{L}\sum_{\substack{k=1\\k\neq j}}^{n} T(\theta - \theta_k),\tag{84}$$

$$T_L^{\mathrm{dr}}(\theta_j, \theta) = -L(\bar{\theta}_j - \theta_j)\rho_L(\theta_j)/\kappa.\tag{85}$$

In terms of them we find

$$T_L^{\mathrm{dr}}(\theta_j, \theta) = T(\theta_j - \theta_n) + \frac{1}{L}\sum_{\substack{k=1\\k\neq j}}^{n} \rho_L(\theta_k)T(\theta_j - \theta_k)T_L^{\mathrm{dr}}(\theta_k, \theta).\tag{86}$$

In the thermodynamic limit, $\rho_L(\theta)$ and $T_L^{\mathrm{dr}}(\theta_j, \theta)$ approach the standard TBA functions $\rho_s(\theta)$ and the dressed differential scattering kernel $T^{\mathrm{dr}}(\theta_j, \theta)$.

## B Computation of the term $C^2$

We consider the contributions from graphs containing two edges. The three types of the leading contributions are shown in Fig. 2. They are all of order $L^4$. The contributions from graphs with an edge originating at the $n+1$-th node are suppressed by $L^2$. In summing over graphs we need to be careful to include only tree-like graphs. It is convenient to actually sum over all the graphs and at the end subtract contributions from graphs containing loops. The 3 leading contributions are

$$C^{2a} = \frac{(2\pi)^2}{2}\sum_{\substack{i,j=1\\i\neq j}}^{n} \frac{f_c^{\mathcal{O}}(\hat{\theta}_i, \hat{\theta}_j)}{\rho_n(\theta_1, \ldots, \theta_n)}\frac{1}{\alpha_i\alpha_j}T(\theta_i - \theta)T(\theta_j - \theta),\tag{87}$$

$$C^{2b} = (2\pi)^2\sum_{\substack{i,j,k=1\\i\neq j,k\neq j}}^{n} \frac{f_c^{\mathcal{O}}(\hat{\theta}_i, \hat{\theta}_j)}{\rho_n(\theta_1, \ldots, \theta_n)}\frac{\alpha_k}{\alpha_i\alpha_j}T(\theta_i - \theta)T(\theta_j - \theta_k),\tag{88}$$

$$C^{2c} = \frac{(2\pi)^2}{2}\sum_{\substack{i,j,k,l=1\\i\neq j,k\neq j,l\neq i}}^{n} \frac{f_c^{\mathcal{O}}(\hat{\theta}_i, \hat{\theta}_j)}{\rho_n(\theta_1, \ldots, \theta_n)}\frac{\alpha_k\alpha_l}{\alpha_i\alpha_j}T(\theta_i - \theta_l)T(\theta_j - \theta_k).\tag{89}$$

There is only one way the loop can be formed with two edges, this corresponds to contribution $C^{2c}$ with $l = j$ and $k = i$. Therefore, the additional contributions that we need to subtract are

$$\bar{C}^2 = \frac{(2\pi)^2}{2}\sum_{\substack{i,j=1\\i\neq j}}^{n} \frac{f_c^{\mathcal{O}}(\hat{\theta}_i, \hat{\theta}_j)}{\rho_n(\theta_1, \ldots, \theta_n)}T^2(\theta_i - \theta_j).\tag{90}$$

We analyze the resulting contributions in few steps. First let us consider the contributions from the first two classes of graphs. We find

$$
\begin{aligned}
C^{2a} + \frac{1}{2}C^{2b} &= \frac{(2\pi)^2}{2} \sum_{\substack{i,j=1 \\ i \neq j}}^{n} \frac{f_c^{\mathcal{O}}(\hat{\theta}_i, \hat{\theta}_j)}{\rho_n(\theta_1, \ldots, \theta_n)} \frac{T(\theta_i - \theta)}{\alpha_i \alpha_j} \left( T(\theta_j - \theta) + \sum_{\substack{k=1 \\ k \neq j}}^{n} \alpha_k T(\theta_j - \theta_k) \right) \\
&= \frac{(2\pi)^2}{2} L \sum_{\substack{i,j=1 \\ i \neq j}}^{n} \frac{f_c^{\mathcal{O}}(\hat{\theta}_i, \hat{\theta}_j)}{\rho_n(\theta_1, \ldots, \theta_n)} \frac{\rho_L(\theta_j) T(\theta_i - \theta)}{\alpha_i}.
\end{aligned}
\tag{91}
$$

In a similar fashion, looking at the second and third class together, we find

$$
\frac{1}{2}C^{2b} + C^{2c} = \frac{(2\pi)^2 L}{2} \sum_{\substack{i,j=1 \\ i \neq j}}^{n} \frac{f_c^{\mathcal{O}}(\hat{\theta}_i, \hat{\theta}_j)}{\rho_n(\theta_1, \ldots, \theta_n)} \frac{\rho_L(\theta_j)}{\alpha_i} \sum_{\substack{k=1 \\ k \neq i}}^{n} \alpha_k T(\theta_i - \theta_k).
\tag{92}
$$

The total contribution $C^2$, with the extra terms subtracted, is then

$$
C^2 = \frac{(2\pi L)^2}{2} \sum_{\substack{i,j=1 \\ i \neq j}}^{n} \frac{f_c^{\mathcal{O}}(\hat{\theta}_i, \hat{\theta}_j)}{\rho_n(\theta_1, \ldots, \theta_n)} \left( \rho_L(\theta_i) \rho_L(\theta_j) - \frac{1}{L^2} T^2(\theta_i - \theta_j) \right).
\tag{93}
$$

Consider now a determinant of the sub-matrix defined in terms of the Jacobian $J_{kl}$ by selecting only the $i$-th and $j$-th columns and rows

$$
\tilde{\rho}_2(\theta_i, \theta_j) = (2\pi L)^2 \det \begin{pmatrix} \rho_L(\theta_i) & -\frac{1}{L} T(\theta_i - \theta_j) \\ -\frac{1}{L} T(\theta_i - \theta_j) & \rho_L(\theta_j) \end{pmatrix}.
\tag{94}
$$

Evaluating the determinant we get

$$
\tilde{\rho}_2(\theta_i, \theta_j) = (2\pi L)^2 \left( \rho_L(\theta_i) \rho_L(\theta_j) - \frac{1}{L^2} T^2(\theta_i - \theta_j) \right).
\tag{95}
$$

Therefore

$$
C^2 = \frac{1}{2} \sum_{\substack{i,j=1 \\ i \neq j}}^{n} \frac{f_c^{\mathcal{O}}(\hat{\theta}_i, \hat{\theta}_j) \tilde{\rho}_2(\theta_i, \theta_j)}{\rho_n(\theta_1, \ldots, \theta_n)}.
\tag{96}
$$

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
