# Peer review of "Generalized hydrodynamics regime from the thermodynamic bootstrap program"

_SciPost Physics, doi:SciPost Phys. 8, 004 (2020)_

## Round 2 · Referee Report · Anonymous · 2019-10-28

Strengths

1- Original computation, based on the previous work by the same authors, which might open the way for many developments in the field of out-of-equilibrium integrable QFT.

2- The paper is clear and well-written.

3- Recent results from Generalized Hydrodynamics are reproduced with success.

Weaknesses

1- A "modified LeClair-Mussardo" formula (25) is used, with a certain regularization scheme for the form factors singularities, however the results presented here at Euler scale do not allow to test the effect of this regularization scheme.

2-The computation still relies on some usual GHD assumptions, namely the vanishing of contribution from higher-order form factors at Euler scale or for diffusive contributions respectively (as opposed to a finite contribution resulting from the re-summation of all terms). While such assumptions are reasonable, they prevent the present results from being a completely independent check of GHD.

3- Lacks any form of numerical check.

Report

The authors expand on their previous construction, the thermodynamic bootstrap program (TBP, ref. [1]), to compute the correlation functions of local operators at the Euler scale (which holds where time and space are commensurately sent to infinity) in the case where such theories have inhomogeneous density/energy/charge profiles, a regime nowadays commonly described by the generalized hydrodynamics (GHD).
In contrast with the traditional form factor bootstrap, the TBP formulates axioms for the form factors of local operators over finite energy backgrounds, typically finite-density eigenstates representing finite temperature or generalized Gibbs ensembles (GGE). In the inhomogeneous setting, such finite energy backgrounds vary continuously with space-time.
The Euler scale correlators are expressed from the TBP form factors through an adaptation of the Leclair-Mussardo series (eq. (25) in the paper), and the leading contribution (single particle-hole pairs, two particle-hole pairs and more) are therefore computed. Only the former contribute at Euler scale, and allow comparison with independent GHD results. Further contribution, in turn, are argued to describe diffusive corrections, and more generally a full gradient expansion of the hydrodynamic equations.

This is an overall very good and original paper, which I recommend for publication in SciPost once the following minor points have been addressed.

Requested changes

1- As pointed in the "Weaknesses" section, the notion of representative state is introduced in p. 10, but used already much earlier in the paper. In fact, in eq. (5) the authors use the notation $\langle \ldots \rangle_{\vartheta}$ (instead of the previously used $\langle \ldots \rangle_{\rm GGE}$ with no further explanation. This should be clarified a bit.

2- Perhaps a naive question : in the form factor axioms, in particular eq. (28), one may have expected that the scattering factor $S()$ would need to be replaced by the dressed scattering phase. Could the authors clarify this point ? (not necessarily in the paper if there is a clear justification for that)

3- Misprint on p. 13, 3rd line : "rapidties" instead of "rapidities"

4- Reference [43] lacks an author

  • validity: high
  • significance: high
  • originality: high
  • clarity: top
  • formatting: excellent
  • grammar: excellent

Author:  Milosz Panfil  on 2019-11-28  [id 658]

(in reply to Report 1 on 2019-10-28)

We would like to thank the referee for their helpful suggestions and analysis of our manuscript. We will briefly comment on the set of Weaknesses pointed out by the referee.

While it is true that at the Euler scale, the regularization scheme for the poles of the Leclair-Mussardo formula is not verified (since the one-particle-hole pair form factor do not generally contain annihilation poles), the regularization scheme does indeed play a role for the computation of the diffusion matrix, using the two-particle-hole pairs form factors. This calculation was done in [30], where using the same regularization prescription, leads to their expression for the diffusion matrix. Since it is a lengthy calculation which has already been done, we prefer to refer the reader to [30], rather than repeat the calculation in our paper.

  1. The previous GHD results for correlation functions in [13] actually had nothing to do with a form factor expansion, and are derived from semiclassical kinematical arguments. From the GHD results alone, it is not possible to know what does this correlation function correspond to in terms of form factors, GHD doesn’t tell you what kind of approximation this is in the form factor expansion. What we have done is to show that the same calculation can be done using the form factor logic alone without GHD assumptions, and we show that the leading one particle-hole pair reproduces fully the GHD prediction. This is the first time this is shown for generic operators in QFT, and it was not previously possible to show this within the GHD formalism alone. In our case it is not an assumption, we are plainly computing the leading term in a form factor expansion, as further corrections are suppressed by higher powers of 1/t, we did not select to compute the one-particle-hole pair for any particular purpose, it is just the leading term. Note for instance the discussion surrounding Eq. 3.41 in Ref [13], where it is conjectured that this correlation function probably arises from the one-particle-hole pair in a form factor expansion, however it was not possible to prove this at that point which is what we have done in this paper.

  2. Since the formula for the two-point function we are deriving is not new, the formula had already been presented in [13], we felt that we did not need to show any further numerical (or otherwise) validation of this formula than was already presented in the original reference. We present only an alternative derivation of this result, but the result for the correlation function is not new. One interesting self consistency check shown in [13] is that the Leclair-Mussardo formula reproduces easily the expected two-point functions for conserved charge density operators, but we felt it would be best to not repeat those checks in this paper.

We now briefly comment on the requested changes:

  1. We agree the notation was somewhat confusing, and it has been changed.

  2. The question of whether one needs to use the dressed S matrix can be seen as a matter of different conventions. Using dressed S-matrices amounts to multiplying the bare S matrix with factors of our dressing function $R_{\vartheta}(\theta_n\vert \theta_1,\dots,\theta_n)$ from Eq. (33), which would account for the change in the scattering phase arising from a particle moving around the finite volume. If we chose to dress the S-matrix, then we would not need to include the R factor in Eqs. 30 and 31. The physical solutions to the form factor equations are the same, whether we include the factors of R in the dressed S-matrices, or whether we use only bare dressed matrices, and include an overall R factor in the periodicity and annihilation pole axioms, so it seems to be a matter of convention where to include this factor in the equations. We chose this convention as it is the one that arises naturally when one derives the form factors as we have done in [1], by using the finite-volume regularization of regular form factors. Once we have derived the action, however, we could chose to eliminate the R factor in favour of dressed S-matrices, if we wished to do so, the physical form factors being invariant under such a change.

  3. And 4. Thanks to the referee for spotting these mistakes, they have been fixed.

---

## Round 2 · Referee Report · Anonymous · 2019-10-31

Strengths

1- interesting and timely results and good calculations

Report

In this paper, the authors further develop the thermodynamic bootstrap program which they have introduced recently. They evaluate exactly the single particle-hole pair form factor at low momentum in terms of a Leclair-Mussardo series. This, in turn, using bootstrap, allow them to evaluate other form factors where momenta of particles and holes coincide or are near to each other. They show that for two particle-hole pairs, this reproduces the conjecture of [30] that led to the calculation of hydrodynamic diffusion in integrable models. Further, they evaluate the form factor expansion to two-point function in the large-space-time limit and show that their framework reproduces the formula proposed in [13] from hydrodynamic projections / linear response theory.

I think this paper is very interesting, both for showing that the framework proposed in [1] is consistent and powerful, and for establishing results that were conjectured, and results that were obtained from hydrodynamic principles instead of microscopic calculations. The paper is well written, and should be published.

Besides minor comments (see below), I have only two comments which I'd like the authors to address; both should be simple to address, the first involves a bit more calculations but I think it is important to address it in order to clarify the situation.

1- Section 6: the authors mention that the higher-particle form factors do not contribute to the correlator. It would be nice if they could provide a more complete argument for this. In particular, is it indeed {\em not necessary} to perform Euler-scale fluid-cell averages in order to have the correct answer? Might higher-particle terms contribute non-vanishing oscillating terms which would go away just under averaging? If not, it would be good that the authors give a full argument, as the result, as stated, eq 65 and 73, is stronger than that predicted in ref [13], and this would be very helpful as it would clarify some Euler-scale subtleties.

2- Comment end of section 4: this is an interesting comment, which, if I understood well, was something discussed already in [B. Doyon, Finite-temperature form factors: a review, SIGMA 3, 011 (2007)], see section 4.4. There, a connection was discussed in the Ising (free fermion) case, where it is understood that analytic continuation changes the quantisation scheme between "on the circle" and "at finite temperature".

Minor comments:

p5: also mention [B. Doyon and H. Spohn, Drude weight for the Lieb-Liniger Bose gas, SciPost Phys. 3, 039 (2017)]

p6: the logic was more that the form factor structure was assumed, then diffusion matrix found and checked in various ways.

P 14 : epsilon is not defined I think

Ref 43 names

eqs 65, 73: maybe define not as a limit (which is zero because of the $1/t$) but as an asymptotic, or alternatively define as a limit after multiplying by $t$

Requested changes

Address comments 1 and 2 above.

  • validity: high
  • significance: high
  • originality: high
  • clarity: high
  • formatting: perfect
  • grammar: excellent

Author:  Milosz Panfil  on 2019-11-28  [id 659]

(in reply to Report 2 on 2019-10-31)

We thank the referee for their insightful report. We would like to reply to some of the comments of the referee:

  1. We have added a discussion about the higher-order corrections to the Euler scale correlators in Section 6. There are two types of leading corrections to our formula, both of which would contribute only at order $1/t^2$ at late times. The first correction comes from the leading correction to the stationary phase approximation we used to calculate the one-particle hole term. This would include contributions from the one-particle-hole pair form factor at non-zero momentum values, however, these subleading corrections of the stationary phase approximation would come only at order $1/t^2$. The other type of correction comes from the leading term in the stationary phase approximation for the two-particle-hole pairs form factor. In this case, there are four integrals over rapidities to be performed, which when performing a similar stationary phase approximation, means that the leading contribution is again $1/t^2$. Therefore if one is interested in only the leading order of the correlation function, that is, in order $1/t$ contributions, then we only need to consider the one-particle-hole pair form factors at zero momentum.

  2. We have added a comment at the end of Section 4. on the previous discussion regarding the dressing of form factors in the different space-time channels. Indeed this is a result we had missed in our review of the literature, so we thank the referee for pointing it out.

Minor comments: 1. We agree Doyon, Spohn (2017) should have been in the references, and we have corrected this. 2. We clarified our comments on the previous derivation of the diffusion matrix. 3. We agree, and we clarified the notation by adding eq. (48) 4. We added the author name for 43. 5. Indeed, at strictly infinite times the correlator vanishes, so we have changed the reference to “limit” and instead discuss now the “large t behavior”.

---

## Round 2 · Referee Report · Anonymous · 2019-11-4

Strengths

1- the paper extends the axioms of thermodynamic bootstrap program formulated in the original reference [1] providing a systematic derivation of the one particle-hole form factor
2- the extended thermodynamic bootstrap program is validated by comparison with existing generalized hydrodynamics (GHD), both for correlation functions (1 particle-hole form factor) and for the diffusion matrix (2 particle-hole form factors)
3- a general method which leads to arbitrary particle hole form factors is given, which could be useful to write down a systematic gradient expansion beyond GHD

Weaknesses

1- there is no check (even numerical) of all predictions beyond the state-of-the-art

Report

This paper provides further results and confirmations about the thermodynamic bootstrap program, introduced in a recent publication by the same authors. This program aims at giving the minimal set of axioms needed to reconstruct the explicit formulas of form factors of a few excitations on top of a thermodynamic GGE state. This is both a relevant and ambitious direction and I found remarkable that so far, all the checks have been passed.
In particular, in this work, it is showed that the single particle-hole form factor at small momentum does indeed reproduce the known result from generalised hydrodynamcs. Additionally, the 2 particle-hole form factor is consistent with the already known diffusion matrix.

Overall, the paper is well-written, although being this an "axiomatic program", it is important to clarify which hypothesis are done for each derivations. I would recommend its publication once my comments have been addressed.

Requested changes

1- the use of the subscript $\vartheta$ in Eq. (5,7,8) is not clear at that stage of the paper. It is only clear later on, but this should be stated explicitly in the text.
2- After Eq. (27), the expression "dressed form" factors is used. I found it confusing because it is not immediately clear if it is the same quantity introduced in Eq. (24) (which is however defined simply as "form factor") or it involves some further dressing operation.
3- After Eq. (57) is derived, the authors say that it can be added as an additional "axiom" to the bootstrap program. I think it would be useful if the logic and terminology were explained more clearly. Indeed, after the axioms (28-33) have been introduced, one would expect that the further derivations are a consequence of these axioms. But then, how can a new axiom be derived? Is it truly independent on the other ones (as they have been used in its derivation)? Could the authors clarify better the problem with the prefactor?
4- Eq. (59) involves a limit where $\kappa_1, \kappa_2 \to 0$ but the right-hand side depends clearly on the way the limit in two variables is approached. As far as I can see, here the idea is that rather than taking really a limit, the authors are analysing the structure of singularities which allows them to derive Eq. (62). Although these calculations and the notation can be standard within this literature, I think it would be useful to be more explicit. The same questions can be raised about Eq. (64).
5- It is stated that at large $t$, only one particle hole matters n Eq. (66). This should be motivated more clearly, perhaps estimating what is the correction due to a larger number of particle-holes.

  • validity: high
  • significance: top
  • originality: high
  • clarity: good
  • formatting: excellent
  • grammar: excellent

Author:  Milosz Panfil  on 2019-11-28  [id 660]

(in reply to Report 3 on 2019-11-04)

We thank the referee for their report and thoughtful comments on our manuscript. We would like to address the comments of the referee.

Weaknesses:

  1. As we also mentioned in our reply to Referee 1, Since the formula for the two-point function we are deriving is not new, the formula had already been presented in [13], we felt that we did not need to show any further numerical (or otherwise) validation of this formula than was already presented in the original reference. We present only an alternative derivation of this result, but the result for the correlation function is not new. One interesting self consistency check shown in [13] is that the Leclair-Mussardo formula reproduces easily the expected two-point functions for conserved charge density operators, but we felt it would be best to not repeat those checks in this paper.

Requested changes:

  1. Indeed we agree the notation might have been confusing, so we have modified it in the manuscript.
  2. We have added a comment below Eq. (25) explaining what we mean by “dressed” form factors, and hope this clarifies the nomenclature.
  3. We have added a discussion below Eq. (59), which explains the relation of our new result to the previous axioms. Eq. (59) is in fact independent of the previously derived axioms, and it is not possible to derive it using the old axioms alone. We therefore say that we can treat it as an additional axiom on the list, that can be used complementarily to fix the form factors on a thermodynamic background. Indeed, we use the word “axiom” somewhat loosely, as these are equations we derive from other simpler first principles. Instead we mean these are axioms relative to the thermodynamic bootstrap program, that is they can be used as the starting point, from which to derive the thermodynamic form factors.
  4. We have slightly rewritten the relevant parts, which we hope clarifies the order of the limits and our focus on non-analyticities.
  5. As we also replied a similar question to Referee 2, we have added a discussion about the higher-order corrections to the Euler scale correlators in Section 6. There are two types of leading corrections to our formula, both of which would contribute only at order $1/t^2$ at late times. The first correction comes from the leading correction to the stationary phase approximation we used to calculate the one-particle hole term. This would include contributions from the one-particle-hole pair form factor at non-zero momentum values, however, these subleading corrections of the stationary phase approximation would come only at order $1/t^2$. The other type of correction comes from the leading term in the stationary phase approximation for the two-particle-hole pairs form factor. In this case, there are four integrals over rapidities to be performed, which when performing a similar stationary phase approximation, means that the leading contribution is now $1/t^2$. Therefore if one is interested in only the leading order of the correlation function, that is, in order $1/t$ contributions, then we only need to consider the one-particle-hole pair form factors at zero momentum.

---

## Round 3 · Referee Report · Anonymous (Referee 1) · 2019-12-4

Report

I thank the authors for answering my points, and recommend the paper in its current version for publication.

---

## Round 3 · Referee Report · Anonymous (Referee 2) · 2019-12-9

Report

All appropriate changes have been and I think the paper can be published now.

---

## Round 3 · Author Response

Dear Editor,

Please find our resubmission of the manuscript to the SciPost Physics.
We have addressed all the points raised by the referees and added a discussion on the in-cell averaging.

Kind regards,
the authors

---

## Round 3 · List of Changes

The main changes are:

1) we have improved notation in the introduction, as requested by the referee 1 and 3;
2) we have provided computations showing that higher particle form factors do not contribute to the correlator at the Eulerian scale (section 6);
3) we have clarified the terminology (dressed form factors, section 3) and added a discussion how the newly derived formula fits within the bootstrap program (end of section 4);
4) we have clarified the notation in the analysis of non-analyticities of form-factors in section 5;
5) we have added a comment that similarity between finite temperature and finite size form-factors was previously observed for Ising field theory (end of section 4);
6) finally, we have included the in-cell averaging procedure in the derivation of the Eulerian scale correlators in section 6. This procedure is needed to fully recover the Eulerian-scale predictions of the GHD. This step of the derivation was missing in the first submission.

---

## Editorial Decision

published